# Organic Nanobowls Modified Thin Film Composite Membrane for Enhanced Purification Performance toward Different Water Resources

**DOI:** 10.3390/membranes11050350

**Published:** 2021-05-10

**Authors:** Changjin Ou, Sisi Li, Zhongyi Wang, Juan Qin, Qian Wang, Zhipeng Liao, Jiansheng Li

**Affiliations:** 1Nantong Key Laboratory of Intelligent and New Energy Materials, School of Chemistry and Chemical Engineering, Nantong University, Nantong 222100, China; ocj1987@ntu.edu.cn (C.O.); LISISI4408@163.com (S.L.); wzy564872358@163.com (Z.W.); qinjuan880816@ntu.edu.cn (J.Q.); 2School of Materials Science and Chemical Engineering, Xi’an Technological University, Xi’an 710021, China; drifting_leaf@126.com; 3Key Laboratory of New Membrane Materials, Ministry of Industry and Information Technology, School of Environment and Biological Engineering, Nanjing University of Science and Technology, Nanjing 210094, China

**Keywords:** nanofiltration, nanofiller, thin film composite membrane, water purification, antifouling property

## Abstract

The structure and composition of nanofillers have a significant influence on polyamide nanofiltration (NF) membranes. In this work, an asymmetric organic nanobowl containing a concave cavity was synthesized and incorporated into a polyamide layer to prepare thin film nanocomposite (TFN) membranes via an interfacial polymerization process. Benefiting from the hydrophilicity, hollow cavity and charge property of the compatible organic nanobowls, the separation performance of the developed TFN membrane was significantly improved. The corresponding water fluxes increased to 119.44 ± 5.56, 141.82 ± 3.24 and 130.27 ± 2.05 L/(m^2^·h) toward Na_2_SO_4_, MgCl_2_ and NaCl solutions, respectively, with higher rejections, compared with the control thin film composite (TFC) and commercial (CM) membranes. Besides this, the modified TFN membrane presented a satisfying purification performance toward tap water, municipal effluent and heavy metal wastewater. More importantly, a better antifouling property of the TFN membrane than TFC and CM membranes was achieved with the assistance of organic nanobowls. These results indicate that the separation performance of the TFN membrane can be elevated by the incorporation of organic nanobowls.

## 1. Introduction

Water scarcity is pushing the revolution of efficient purification technologies. Membrane separation, including reverse osmosis [1,2], nanofiltration (NF) [3,4,5,6], ultrafiltration [7,8,9], distillation [10,11,12] and forward osmosis [13,14], has been considered as one of the most reliable technologies for water purification [15,16,17,18]. Among these, NF has prevailed in seawater desalination, wastewater treatment and advanced water purification, due to its high selectivity and moderate cost of operation. However, its broader application is stunted by the typical trade-off between permeability and selectivity [19]. Therefore, simultaneously improving the permeability and selectivity of NF membranes to break the trade–off phenomenon is a formidable challenge.

Introducing inorganic nanofillers into the polyamide (PA)–selective layer to prepare the thin film nanocomposite (TFN) membrane is an efficient strategy to achieve this end. The embedded inorganic nanofillers (e.g., carbon, silica and metal-based nanomaterials) are able to regulate membrane properties such as charge, hydrophilicity and PA layer thickness to reach a more favorable state, for the purpose of simultaneously elevating the membrane permeability and selectivity [20,21,22,23]. Nevertheless, these traditional inorganic nanofillers are poorly compatible with the polymeric membrane matrix or PA layer, which probably results in unselective interfacial defects, thus impairing membrane selectivity [24,25,26,27]. Besides this, the suitable dimension of nanofillers is basically determined by the thickness of the membrane PA layer, which generally ranges from 50 to 200 nm [15]. Larger nanofillers might not be beneficial in the modification of membranes, because the oversized nanofillers are likely to cause interfacial defects and leaching risk under the dynamic filtration process [28]. In addition, it has been proven that the internal cavity of nanofillers is conducive to supplying more free volumes to the PA layer, which can reduce the transport resistance of the water molecules and enhance membrane permeability [29,30,31]. For these reasons, organic nanofillers with good compatibility, suitable effective size and internal cavity are highly desired for the development of TFN membranes.

In this study, compatible organic nanobowls with an internal concave cavity and asymmetric structure (lower height–diameter ratio) were synthesized through a modified Stöber method and then embedded into the PA layer via interfacial polymerization for fabricating a novel TFN NF membrane. The basic characteristics of the synthesized organic nanobowls and composite membranes were investigated. Except for the simulated saline, the novelty of this work focuses on the evaluation of separation performances of the TFN membrane using real water resources, including seawater, tap water, municipal effluent and heavy metal wastewater, and a performance comparison with those of the control thin film composite (TFC) and commercial (CM) NF membranes. The effects of developed TFN membrane parameters, such as surface charge, pore size and hydrophilicity, on the separation performance were also discussed. These explorations aimed to indicate that organic nanobowl-modified TFN membranes are superior to TFC and CM NF membranes in purifying different water resources.

## 2. Materials and Methods

### 2.1. Reagents

Polysulfone membrane substrate (PSf, MWCO = 20 kDa) was supplied by RisingSun Membrane Technology Co., Ltd. (Beijing, China) and was applied to the fabrication of NF membranes. NF270 was purchased from Dow Filmtech (Orlando, FL, USA) as CM NF membrane. Piperazine (PIP), ethylenediamine (EDA), formaldehyde, sodium chloride (NaCl), sodium sulfate (Na_2_SO_4_), magnesium chloride (MgCl_2_), tetraethylorthosilicate (TEOS), resorcinol, hydrofluoric acid (HF) and *n*-hexane were provided by Sinopharm Chemical Reagent Co., Ltd. (Nanjing, China). Trimesoyl chloride (TMC) was obtained from Aladdin Co., Ltd. (Shanghai, China). Deionized (DI) water was used in each experimental step and was produced by a Millipore water purification system. The chemicals are analytical grade and used without further purification.

### 2.2. Synthesis of Organic Nanobowls and Composite Membranes

The organic nanobowls were synthesized using a modified Stöber method [32]. First, 1.0 mL EDA was added into a mixed solution of 53.4 mL ethanol and 26.6 mL DI water. After stirring for 30 min, 2.8 mL TEOS was dropped into the above solution and then 0.4 mL resorcinol was added. Subsequently, 0.56 mL formaldehyde was added into the mixed solution before stirring for another 24 h. The centrifugation was further etched with HF (to remove silicons for the formation of the bowl shape of nanobowls) and water-washed until reaching a neutral pH. The obtained orange powder was dried at 80 °C overnight before use.

The control TFC membrane was fabricated on the PSf substrate via interfacial polymerization. Briefly, an aqueous solution containing 0.35 wt% PIP was poured onto the membrane surface and kept for 2 min before the removal of excess solution. Then, the membrane was immersed into an organic solution containing 0.1 wt% TMC for 1 min. Finally, the obtained TFC membrane was heat-treated at 60 °C for 15 min and stored in DI water. For the TFN membrane, the preparation process is the same as that of the TFC membrane, except the aqueous solution was replaced with a solution containing 0.35 wt% PIP and 0.12 wt% organic nanobowls.

### 2.3. Characterization

The surface morphologies of nanobowls and composite membranes were characterized by scanning electron microscope (SEM, JEOL 7800, JEOL, Tokyo, Japan). The structure of nanobowls was confirmed by transmission electron microscope (TEM, T20, FEI, Hillsborough, OR, USA). The surface roughness of composite membranes (5 μm × 5 μm) was measured by atomic force microscopy (AFM, Bruker MultiMode 8, Bruker AXS, Billerica, MA, USA). The surface hydrophilicity and charge property of composite membranes were inspected by a contact angle tester (WCA, DSA30, Krüss, Hamburg, Germany) and a zeta potentiometer (SurPASS 3, Anton Paar, Graz, Austria), respectively. The results were averaged from 3 repeats. The surface element ratios of composite membranes were investigated by X-ray photo electron spectroscopy (XPS, PHI Quantera II, Chigasaki, Japan).

### 2.4. Membrane Performance Test

The separation performances of composite membranes were examined by a lab-assembled cross-flow apparatus. The experiments were carried out using a membrane piece with an effective area of 12.56 cm^2^ at 6.0 bar. The basic separation performances (flux (*J*) and salt rejection (*R*)) of the composite membrane were investigated with 1 g/L salt solutions. The corresponding equations were listed as follows
(1)$$J=\frac{V}{S\times t}$$
(2)$$R=\left(1-\frac{C_{p}}{C_{f}}\right)\times 100\%$$
where *V* is the permeation volume (L), *S* is the effective filtration area of membranes (m^2^), *t* is the separation time (h). *C_p_* and *C_f_* (g/L) correspond to the solute concentrations in the permeation and feeding, respectively. The concentrations of salts in the feed and permeate solutions were tested by a conductivity meter (DDSJ 308A, Leici, Shanghai, China).

The seawater was collected from the coast in Qingdao, Shandong Province, China (location from Google maps: 36°09′27.6″ N 120°21′08.8″ E). The tap water was obtained from the pipeline networks of the water supply system. The municipal effluent and heavy metal wastewater were acquired from the East City sewage plant and industrial park of Nanjing, respectively. In the experiment, the concentrations of cations, such as Na^+^, K^+^, Mg^2+^ and Ca^2+^, in the different water resources, were investigated by inductively coupled plasma spectrometry (ICP-OES, PerkinElmer Optima 7000 DV, Waltham, Massachusetts, USA). The concentrations of anions, such as Cl^−^ and SO_4_^2−^, were analyzed by ion chromatography (Aquion ics-2100, Waltham, MA, USA). The concentration of NO_3_^-^ was recorded using an UV-vis spectrophotometer (Lambda 25, PerkinElmer, Waltham, MA, USA) at a wavelength of 220 nm. The content of organic components in the water samples was surveyed by a total organic carbon (TOC) analyzer (TOC-VCSH, Shimadzu, Tokyo, Japan). Viscosity of the feed and permeated water were tested by a viscometer (DV-II+ Pro, Brookfield, Middleboro, MA, USA) and the speed was set at 20 rpm. The pH values of the feed and permeated water were measured by a pocket pH tester (pH scan 30, Bante, Shanghai, China). Before separating different real water resources, the pH values were adjusted to the suitable range of 5.0–8.0 of NF membranes. After this, the water resources were pre-treated with 0.45 μm polypropylene microfiltration membrane to remove the suspensions and precipitates. In the experiment, the separation performances of all membranes were obtained from three repetitions.

## 3. Results and Discussions

### 3.1. Characterizations of Organic Nanobowls and Composite Membranes

The morphology of organic nanobowls was investigated and presented in Figure 1. It can be observed that the asymmetrical organic nanobowls are uniformly shaped with a concave cavity (Figure 1a). Once organic nanobowls are successfully introduced into the membrane, this internal cavity, providing more free volumes to the PA layer, is beneficial in decreasing the transport resistance of water molecules. Additionally, the pure organic feature is conducive to the absence of unselective interfacial defects between nanofillers and the membrane matrix. Based on the image of dispersion in the aqueous solution (inset of Figure 1a), the good dispersibility of organic nanobowls was ensured by the typical “Tyndall effect”, which is highly desired for its ability to help avoid the agglomeration of nanofillers in PA layer. In addition, the size of the organic nanobowls was 220–350 nm (Figure 1b), indicating a height of about 110–175 nm, which matches the thickness of the PA layer (50–200 nm) [33].

The morphologies of composite membranes were revealed by SEM and AFM. It can be seen from Figure 2a that the control TFC membrane presents as a smooth surface similar to the CM membrane except for some protuberances (Figure 2a,c). Distinctively, owing to the sufficient decoration of organic nanobowls, the TFN membrane is highly rugged (Figure 2b), which can be further proven by the corresponding AFM results (Figure 2e). The rougher surface of the TFN membrane is able to supply more contact sites between feeding water and the selective layer, favoring an enhanced permeability [34,35,36].

Apart from the surface morphology, charge property is another crucial factor that affects the separation performance of membranes, because the charge-based Donnan effect dominates the selectivity of the NF membrane [37]. As presented in Figure 3a, although the entire surface charge of membranes decreases with the increase in pH value, the potential of the TFN membrane is higher than those of the TFC and CM membranes along the test range. These results indicate the higher positive charge of the TFN membrane after the incorporation of organic nanobowls. In addition to the charge property, the pore size of the membrane selective layer is also important for the separation performance. The pore size of the NF membrane can be mirrored by its surface elemental composition. Generally, higher O/N and C/N ratios represent a lower crosslinking degree or larger pore size of the membrane [38,39]. It can be observed from Figure 3b and Table 1 that the O/N and C/N ratios of the TFN membrane are the highest among the three membranes, which demonstrates that the TFN membrane has the largest pores. This result is induced by the changed IP process due to the interference of organic nanobowls. It can also be observed from Figure 3b that the peak corresponding to silicon is absent, which is ascribed to the removal of etching by HF. The membrane hydrophilicity is also presented in Table 1. The water contact angle of the TFN membrane is lower than that of the TFC membrane, which demonstrates the better hydrophilicity of the TFN membrane. The enhanced hydrophilicity of the TFN membrane mainly owes to the exposure of the hydrophilic organic nanobowls of the PA layer.

### 3.2. Membrane Performance Evaluation

The basic nanofiltration performances of developed membranes were investigated using 1 g/L Na_2_SO_4_, MgCl_2_ and NaCl solutions. As shown in Figure 4a, the rejection of the TFN membrane toward Na_2_SO_4_ is comparable to those of the TFC and CM membranes, which indicates the avoidance of unselective interfacial defects thanks to the good compatibility of organic nanobowls. In comparison, the MgCl_2_ rejection of the TFN membrane is 5.97% and 2.16% higher than those of the TFC and CM membranes, respectively. This difference is mainly due to the weaker negative charge of the TFN membrane, which reduces the electrostatic adsorption force between the negatively charged membrane surface and positively charged salts [40]. However, it can also be observed from Figure 4a that the NaCl rejection of TFN membrane was the lowest among the three membranes. This result is explained by the largest pore size of the PA layer of TFN membrane (as demonstrated by the XPS results), which allows the facile passage of electroneutral NaCl. The water fluxes of the composite membranes are presented in Figure 4b. Obviously, owing to the better hydrophilicity (favoring the membrane wettability), rougher surface (providing more effective contact sites) and more free volumes (induced by the organic nanobowls, decreasing mass transport resistance), the TFN membrane shows the highest water flux of 119.44 ± 5.56, 141.82 ± 3.24 and 130.27 ± 2.05 (L/(m^2^·h)) LHM toward Na_2_SO_4_, MgCl_2_ and NaCl solutions, respectively, compared with the TFC and CM membranes.

To further investigate the separation performance of the composite membranes, various water resources, e.g., seawater, tap water, municipal effluent and heavy metal wastewater, were applied. The corresponding water fluxes of TFC, TFN and CM membranes toward seawater are 69.89 ± 2.72, 102.68 ± 4.39 and 44.36 ± 1.66 LHM, respectively. Clearly, the TFN membrane presented the highest flux, which was ascribed to its elevated hydrophilicity, effective contact sites and free volumes of the selective PA layer. As shown in Figure 5 and Appendix A, all composite membranes are able to effectively remove ions and organic species from seawater. Notably, the rejections of SO_4_^2−^ and TOC of all membranes are around 80%. However, compared with the TFC and CM membranes, the TFN membrane shows higher rejection rates for Mg^2+^ and Ca^2+^ cations. These results are induced by the dominant Donnan effect of the TFN membrane based on the absence of unselective interfacial defects [41]. The relatively lower negative charge of the TFN membrane is beneficial for weakening the electrostatic adsorption between the membrane surface and the cations, leading to elevated cation rejection. Besides, the rejection of the TFN membrane toward monovalent ions with smaller size and weaker charge property is lower than those of the TFC and CM membranes, which is due to its looser selective PA layer, reducing the mass transport resistance. In addition, the looser structure of the TFN membrane also causes the slightly decreased removal of organic species, as demonstrated by the TOC results in Appendix A.

Except for desalination, the NF membranes are also qualified for advanced water treatment [42,43]. In this study, tap water and municipal effluent were applied to explore the membranes’ separation performance. It can be observed from Figure 6a that the fluxes of the membranes toward tap water are higher than that toward the municipal effluent. This might be owing to fouling of the membrane caused by the municipal effluent having more ions and organic components compared with the tap water. In spite of this, the TFN membrane presents a higher water flux toward both of the tap water and municipal effluent than the TFC and CM membranes. From Figure 6b,c, Appendix A, it can easily be seen that the TFN membrane is more competent in Ca^2+^ and Mg^2+^ removal. Meanwhile, the comparable TOC removal (~85%) of tap water and municipal effluent also supports the satisfactory water purification capacity of the TFN membrane.

Further, the NF membranes prevailed in treating industrial wastewater including heavy metal wastewater [44,45,46,47,48]. Before separation, the pH of the heavy metal wastewater was adjusted to a suitable working condition for the NF membrane followed by pre-filtration with a microfiltration membrane. It can be seen in Figure 7a that the fluxes in all membranes are in the order of Ni^2+^ > Zn^2+^ > Cr^3+^. This was caused by the higher concentration of ions (Ni^2 +^< Zn^2+^ < Cr^3+^) (Appendix A), which resulted in higher osmotic pressure across the membrane, leading to a decreased water flux. However, the fluxes in the TFN membranes are superior to the other membranes thanks to their better hydrophilicity, higher number of separation sites and free volumes, as discussed above.

The removal capacity against various heavy metal ions of developed membranes is displayed in Figure 7. Apparently, all membranes possess a high rejection of more than 90% toward three kinds of heavy metals. Besides this, among the selected heavy metals, due to their larger size and stronger charge, the rejections of all membranes toward Cr^3+^ are higher than those of the Ni^2+^ and Zn^2+^. Specifically, the rejections of TFC, TFN and CM toward Cr^3+^ are 98.62 ± 0.19%, 99.35 ± 0.24% and 99.13 ± 0.13%, respectively. These satisfying results also can be mirrored by the color change of their permeation (Appendix A). Despite this, the TFN membrane is preferred to the TFC and CM membranes, because the rejections of the three heavy metals were the highest at more than 98.50%. The satisfying rejection of the TFN membrane is supported by the good compatibility between the organic nanobowls and polymeric matrix, which avoids the presence of interfacial defects [49,50,51]. Fundamentally, since the selectivity of the NF membrane is significantly influenced by the Donnan effect, the achieved distinctive performance of the TFN membrane can mainly be attributed to the ameliorated surface charge property.

Similar to the above results, except for the heavy metals, the TFN membrane also presents a higher rejection to Ca^2+^, Mg^2+^ and Cl^-^ and a comparable rejection to the SO_4_^2-^ in the heavy wastewater. In addition, the pH values of the different water resources are slightly increased, which might be due to the changes in the dissolved ions in the solutions. Meanwhile, the TFN membranes are able to effectively remove organic species. Taking Zn^2+^ heavy metal wastewater as an example, the TOC values decreased from 37.1 ± 1.28 to 6.21 ± 1.70 mg/L, realizing an 83.25% removal rate. More importantly, benefiting from the stronger hydrophilia and weaker negative charge, the fouling extent of the TFN membrane is much better than the TFC and CM membranes (Appendix A), which is highly desired for the purification of practical water resources.

Despite the excellent performance of the TFN membrane, the concentrations of heavy metals of Ni^2+^ (9.08 ± 3.82 mg/L), Cr^3+^ (8.23 ± 3.04 mg/L) and Zn^2+^ (14.91 ± 3.93 mg/L) in the permeation are still higher than the National Emission Standards GB8978-1996 and GB25466-2010 (Ni^2+^ of 1.00 mg/L, Cr^3+^ of 1.50 mg/L and Zn^2+^ of 1.50 mg/L), due to the excessive concentrations of the feed solutions. Therefore, second-stage treatment with NF membranes was carried out for further purification. As measured, the concentrations of Ni^2+^ and Zn^2+^ in the secondary permeation decreased to 0.27 ± 0.09 and 0.30 ± 0.04 mg/L, respectively, while the concentration of Cr^3+^ was lower than the detection limit, which indicates the qualified treatment of heavy metal wastewater by the TFN membrane.

## 4. Conclusions

In this work, organic nanobowls were synthesized and used as nanofillers for preparing high-performing TFN nanofiltration membranes. The compatible organic nanobowls modified TFN membrane, which proved superior to the control TFC and CM membranes due to its higher flux and salt rejection due to enhanced surface hydrophilia and more effective separation sites and free volumes, as well as the ameliorated charge property of the selective PA layer. Besides this, the TFN membrane is capable of effectively removing dissolved cations and organic species in different real-water resources. More importantly, the TFN membrane presented higher permeability and selectivity toward selected heavy metal wastewater resources than those of the TFC and CM membranes. Meanwhile, a lower fouling extent of the TFN membrane surface was obtained during separation. In brief, this work provides a new strategy to prepare adaptive TFN nanofiltration membranes for the purification of different water resources.

## Figures and Tables

**Figure 1 membranes-11-00350-f001:**
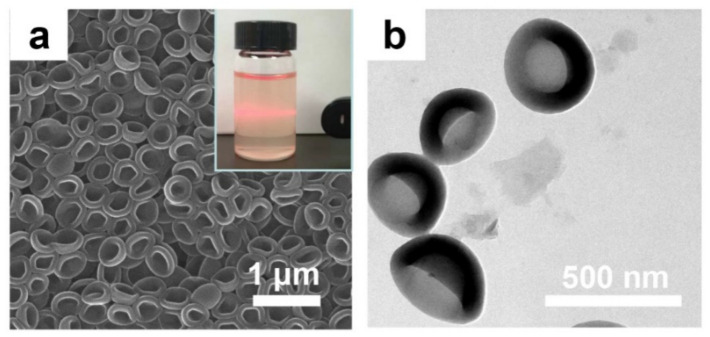
(**a**) SEM and (**b**) TEM images of organic nanobowls. The inset is a photo of organic nanobowl solution.

**Figure 2 membranes-11-00350-f002:**
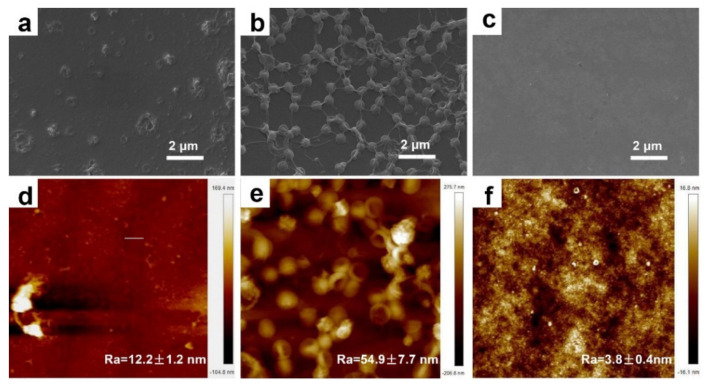
(**a**–**c**) and (**d**–**f**) are the SEM and 2D AFM images of TFC, TFN and CM membranes, respectively.

**Figure 3 membranes-11-00350-f003:**
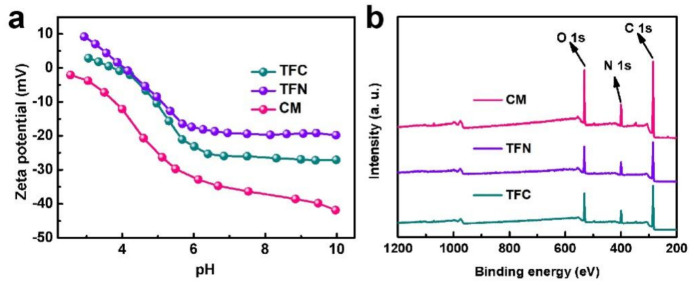
(**a**) Zeta potentials and (**b**) XPS results of the composite membranes.

**Figure 4 membranes-11-00350-f004:**
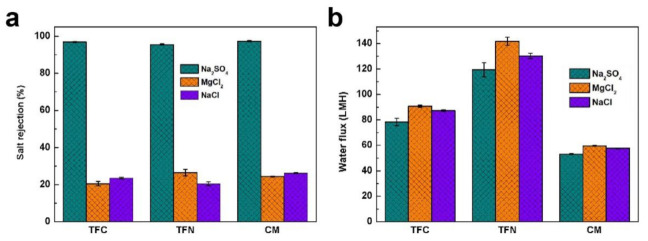
(**a**) Salt rejections and (**b**) water fluxes of the composite membranes.

**Figure 5 membranes-11-00350-f005:**
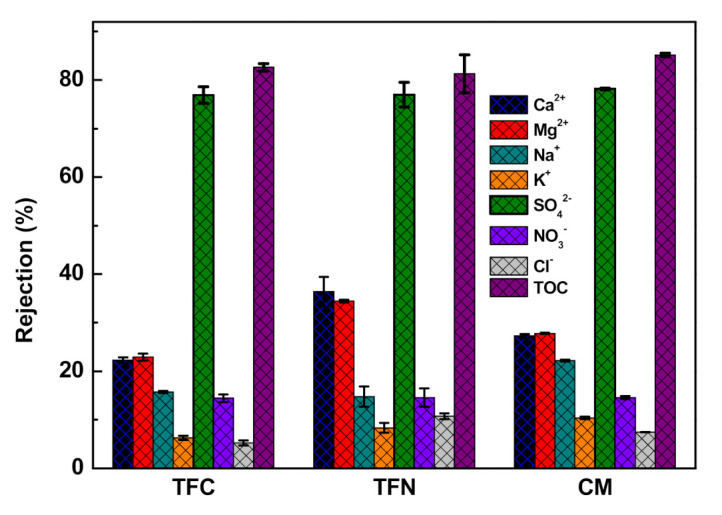
Rejections of the composite membranes toward seawater.

**Figure 6 membranes-11-00350-f006:**
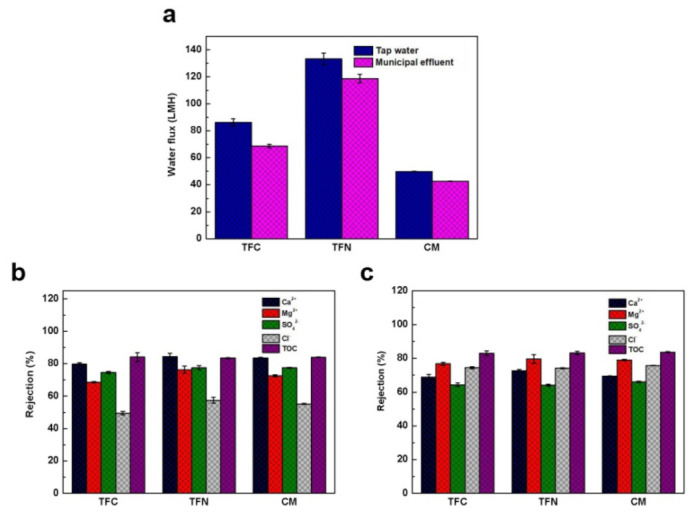
(**a**) Water fluxes of the composite membranes toward tap water and municipal effluent. Rejections of the composite membranes toward (**b**) tap water (**c**) municipal effluent.

**Figure 7 membranes-11-00350-f007:**
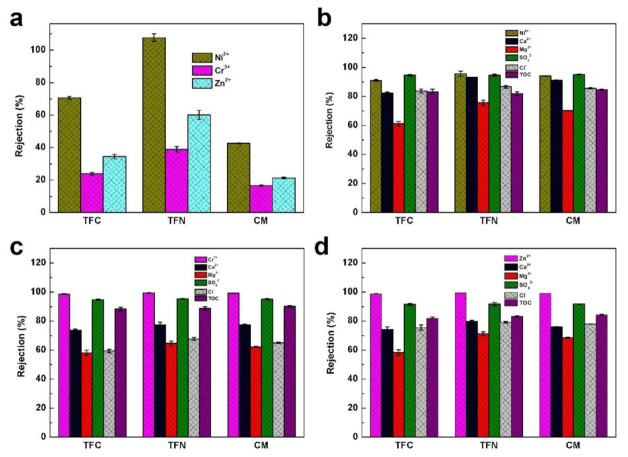
(**a**) Water fluxes of the composite membranes toward heavy metal wastewater. Rejections of the composite membranes toward (**b**) Ni^2+^, (**c**) Cr^3+^ and (**d**) Zn^2+^ wastewater.

**Table 1 membranes-11-00350-t001:** Surface elemental composition of the composite membranes.

Membrane Type	C (%)	N (%)	O (%)	C/N	O/N	WCA (°)
TFC	70.94	10.87	18.19	6.52	1.67	55.2 ± 2.7
TFN	70.28	11.02	18.70	6.38	1.70	42.3 ± 4.6
CM	71.08	11.38	17.55	6.24	1.54	28.5 ± 0.7

## Data Availability

The data presented in this study are available on request from the corresponding author.

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
