# Peer review of "Organic Nanobowls Modified Thin Film Composite Membrane for Enhanced Purification Performance toward Different Water Resources"

_membranes, 2021, doi:10.3390/membranes11050350_

Round 1

Reviewer 1 Report

The manuscript reports using nanobowls-incorporation to improve the water purification performance. However, the prepared TFN didn't show significant improvement in performance, mostly within 1-2%. Therefore, these scientific  results could not really conclude the novelty and excellence of the role of nanobowls. More and further investigations are needed to understand the mechanism.

Author Response

Comments

The manuscript reports using nanobowls-incorporation to improve the water purification performance. However, the prepared TFN didn't show significant improvement in performance, mostly within 1-2%. Therefore, these scientific results could not really conclude the novelty and excellence of the role of nanobowls. More and further investigations are needed to understand the mechanism.

Response

We are sorry for the negative impression from the Reviewer 1 on our manuscript. The separation performance of membranes mainly relies on the water flux and rejection. In this manuscript, we firstly proved the flux increments of more than 50% and 120% of our TFN membrane than those of the TFC and CM membranes, respectively, toward simulated saline. Meanwhile, the MgCl2 rejection of developed TFN membrane is 5.6% and 2.1% higher than those of the TFC and CM membranes, respectively. Moreover, the enhanced separation performance of our TFN membrane was further validated using different real water resources. Based on these satisfying results, the fluxes and rejections of our TFN membrane are much better than those of the TFC and CM membranes. In addition, the increases of rejections of our TFN membranes are not as low as 1-2% (the data was reorganized in Figure 4-7 and Supporting Information). For instance, as referred in the Figure 6 and Table S2, the rejection of developed TFN membrane toward Mg2+ in tap water is 7.85% and 3.87% higher than those of the TFC and CM membranes, respectively. Besides, the separation mechanism of the membranes largely depends on their sieving pore size and electrostatic interaction, which has been mentioned in the manuscript for several times. These statements have been marked in yellow for your reference.

Page 5, “This difference is mainly due to the weaker negative charge of TFN membrane, which weakens the electrostatic adsorption force between the negatively charged membrane surface and positively charged salts [40].

Page 5, “This result is explained by the largest pore size of the PA layer of TFN membrane (as demonstrated by the XPS results), which allows the facile passage of electroneutral NaCl.

Page 6, “The relatively lower negative charge of the TFN membrane is beneficial for weakening the electrostatic adsorption between the membrane surface and the cations, leading to an elevated cation rejection.

Page 7, “Fundamentally, since the selectivity of the NF membrane is significantly influenced by the Donnan effect, the achieved distinctive performance of the TFN membrane is mainly attributed to the ameliorated surface charge property.

Page 8, “More importantly, benefiting from the stronger hydrophilic and weaker negative charge, the fouling extent of the TFN membrane is much better than the TFC and CM membranes (Fig. S2), which is highly desired for the purification of practical water resources.

Reviewer 2 Report

The paper by Li et al. details a method to synthesise a novel membrane material based on bowl-like nanostructures. The study is scientifically sound and should be accepted for publication in Membranes after minor revisions:

- In the reagents section, please provide purities/gardes of chemicals to enhence reproducibility of your methods.

- XPS is used as effectively an elemental analysis tool (which is perfectly fine)- I was curious as to the absence of silicon as TEOS is used in the prep- please can you comment on this? Has the Si been etched away entirely by the HF treatment?

- On page 5, the authors said: "However, it can also be observed from Fig. 4a that the NaCl rejection of TFN membrane was the lowest among the three membranes. This result is explained by the largest pore size of the PA layer of TFN membrane (as demonstrated by the XPS results), which allows the facile passage of electroneutral NaCl." - Surely you mean from the WCA results? you can't  tell anything about pore size from XPS...

- Figure 2 is a microscopy composite figure, however the images are lacking scale bars (as the authors have done in Figure 1). Please add this information.

- In Figures 4, 5, 6 and 7, how many times were the experiments repeated?
 Ion removal studies should be repeated 3-5 times and error bars added to graphs to improve accuracy and reproducibility.

- Were there any kinetic studies performed on the removal of metal ions?

Author Response

General comments:

The paper by Li et al. details a method to synthesise a novel membrane material based on bowl-like nanostructures. The study is scientifically sound and should be accepted for publication in Membranes after minor revisions:

Response

We thank Reviewer 2 for the positive comments. Following Referee 2’s suggestion, the manuscript has been carefully revised accordingly.

Comment 1

- In the reagents section, please provide purities/gardes of chemicals to enhance reproducibility of your methods.

Response

We thank the Reviewer 2 for the helpful comments. Following Reviewer 2’ suggestion, the purities of the reagents were supplied in the revision. The corresponding revision is listed as follows:

Page 2, “The chemicals are analytical grade and used without further purification.

Comment 2

- XPS is used as effectively an elemental analysis tool (which is perfectly fine)- I was curious as to the absence of silicon as TEOS is used in the prep- please can you comment on this? Has the Si been etched away entirely by the HF treatment?

Response

We thank the Reviewer 2 for the thoughtful comments. Just as mentioned by the Reviewer 2, HF is applied to etch the internal silicon cores of the precursors. After the removal of silicon, the spherical precursors collapse to the bowl shapes. The corresponding revision is listed as follows:

Page 2, “The centrifugation was further etched with HF (to remove silicons for the formation of bowl shape of nanobowls) and water-washed until reaching neutral pH. ”

Page 5, “It can also observed from Fig. 3b that the peak corresponding to silicon is absent, which is ascribed to the etching removal by HF.

Comment 3

- On page 5, the authors said: "However, it can also be observed from Fig. 4a that the NaCl rejection of TFN membrane was the lowest among the three membranes. This result is explained by the largest pore size of the PA layer of TFN membrane (as demonstrated by the XPS results), which allows the facile passage of electroneutral NaCl." - Surely you mean from the WCA results? you can't tell anything about pore size from XPS...

Response

We thank the Referee 2 for the useful comments. The rejection of the nanofiltration membrane is significantly influenced by its polyamide (PA) selective layer, which formed by the crosslinking between amino (N-H) and acyl chloride (CO-Cl) groups. Generally, the higher C/N and O/N ratios indicate the lower crosslinking degree or loser structure of PA, thus leading to a larger pore size. Among the three membranes, the C/N and O/N ratios of the TFN membrane are the highest. Whereas the WCA is the indication of membrane hydrophilicity, which has a great impact on the membrane permeance.

Comment 4

- Figure 2 is a microscopy composite figure, however the images are lacking scale bars (as the authors have done in Figure 1). Please add this information.

Response

We are sorry for the absence of the scale bars in the SEM results. Following Reviewer 2’ suggestion, the corresponding scale bars are appended in the revisions. The corresponding revision is listed as follows:

Page 4,

Figure 2. (a) to (c) and (d) to (f) are the SEM and 2D AFM images of TFC, TFN and CM membranes, respectively.

Comment 5

- In Figures 4, 5, 6 and 7, how many times were the experiments repeated? Ion removal studies should be repeated 3-5 times and error bars added to graphs to improve accuracy and reproducibility.

Response

We thank the Referee 2 for the helpful comments. Following Referee 2’ suggestion, the repeated experiments were conducted and corresponding error bars of Figure 4-7 were supplied in the revision. The corresponding revision is listed as follows:

Page 3, “In the experiment, the separation performances of all membranes were obtained from three repetitions.

Page 6,

Figure 4. (a) Salt rejections and (b) water fluxes of the composite membranes.

Figure 5. Rejections of the composite membranes toward seawater.

Page 7,

Figure 6. (a) Water fluxes of the composite membranes toward tap water and municipal effluent. Rejections of the composite membranes toward (b) tap water (c) municipal effluent.

Page 8,

Figure 7. (a) Water fluxes of the composite membranes toward heavy metal wastewater. Rejections of the composite membranes toward (b) Ni2+, (c) Cr3+ and (d) Zn2+ wastewater.

Comment 6

- Were there any kinetic studies performed on the removal of metal ions?

Response

We thank the Referee 2 for the constructive comments. We agree with the Referee 2 on the kinetic studies. However, the feed solutions in this manuscript are real water resources containing complex contaminants, which is unbeneficial for the precise investigation of kinetic studies. This work is devoted to revealing the practical application of our TFN membranes for purifying different water resources. In the following works, we will investigate the separation kinetic studies of the membrane toward the suitable target water resources.

Reviewer 3 Report

Comments to the Author:

The manuscript reports nanobowl-modified thin film composite membrane and how it compares to non-modified and control membranes in water purification. The author presented detailed experiments and analysis to prove that incorporation of nanobowl as filler improves purification performance.

The experiments are well planned and the manuscript is generally well written. And I recommend publication after consideration of the following points:

  1. Since the differences of TFN and the controls in the membrane performance studies are relatively small, have the authors done statistic analysis to show triplicate experiments were considered and/or there was a statistically significant difference between the TFN and the control membranes?
  2. The authors should consider emphasizing the novelty of the current work in this manuscript, especially since they have reported the same nanobowl-modified membrane in previous publications.

Author Response

General comments:

The manuscript reports nanobowl-modified thin film composite membrane and how it compares to non-modified and control membranes in water purification. The author presented detailed experiments and analysis to prove that incorporation of nanobowl as filler improves purification performance. The experiments are well planned and the manuscript is generally well written. And I recommend publication after consideration of the following points.

Comment 1

Since the differences of TFN and the controls in the membrane performance studies are relatively small, have the authors done statistic analysis to show triplicate experiments were considered and/or there was a statistically significant difference between the TFN and the control membranes?

Response

We thank the Referee 3 for the useful comments. Following Referee 2’ suggestion, the repeated experiments were conducted and corresponding error bars of Figure 4-7 were supplied in the revision. The corresponding revision is listed as follows:

Page 3, “In the experiment, the separation performances of all membranes were obtained from three repetitions.

Page 6,

Figure 4. (a) Salt rejections and (b) water fluxes of the composite membranes.

Figure 5. Rejections of the composite membranes toward seawater.

Page 7,

Figure 6. (a) Water fluxes of the composite membranes toward tap water and municipal effluent. Rejections of the composite membranes toward (b) tap water (c) municipal effluent.

Page 8,

Figure 7. (a) Water fluxes of the composite membranes toward heavy metal wastewater. Rejections of the composite membranes toward (b) Ni2+, (c) Cr3+ and (d) Zn2+ wastewater.

Comment 2

The authors should consider emphasizing the novelty of the current work in this manuscript, especially since they have reported the same nanobowl-modified membrane in previous publications.

Response

We thank the Referee 3 for the helpful comments. Our previous work focused on the effects of the composition and structure of nanofillers on the membrane separation performance. Since the separation performances of TFN membrane toward real water resources were seldom investigated (most of previous reports used simulated wastewater), in this work, our TFN membranes were applied to validate the separation performance using different real water resources including tap water, municipal effluent and heavy metal wastewater. Besides, the influences of membrane characters (e.g., charge, pore size and hydrophilicity) on the separation performances were discussed. The corresponding revision is listed as follows:

Page 2, “Except for the simulated saline, the novelty of this work focuses on the evaluation of separation performances of the TFN membrane using real water resources, including seawater, tap water, municipal effluent and heavy metal wastewater, and performances comparison with those of the control thin film composite (TFC) and commercial (CM) NF membranes. Besides, the effects of developed TFN membrane parameters, such as surface charge, pore size and hydrophilicity, on the separation performances were discussed.

Round 2

Reviewer 1 Report

Authors have addressed reviewers' comments. Therefore, I recommend it to be accepted after proper English editing.